# University Fairness Questionnaire (UFair): Development and Validation of a German Questionnaire to Assess University Justice—A Study Protocol of a Mixed Methods Study

**DOI:** 10.3390/ijerph192316340

**Published:** 2022-12-06

**Authors:** Raphael M. Herr, Veronika M. Deyerl, Jennifer Hilger-Kolb, Katharina Diehl

**Affiliations:** 1Department of Medical Informatics, Biometry and Epidemiology, Friedrich-Alexander-Universität Erlangen-Nürnberg (FAU), 91054 Erlangen, Germany; 2NAKO e.V., 69123 Heidelberg, Germany

**Keywords:** students, stress, health, fairness, justice, questionnaire, development, validation, mixed-methods, university

## Abstract

Distress is a widespread phenomenon in the general population, but also among university students, associated with poorer learning success and negative health consequences. A source of distress might be the experience of injustice. Theoretical and empirical work in the area of perceived fairness in the workplace (“organizational justice”) has shown that perceived unfairness is related to various stress indicators and health outcomes. Preliminary evidence indicates that unfairness matters not only in the work context but also in the university context. However, an adapted and validated tool to assess perceived unfairness in the university context is hitherto missing. The goal of the proposed project is therefore to adapt the construct of organizational justice to the university context and to develop a corresponding questionnaire by means of established scientific procedures. An exploratory sequential mixed-methods design is applied in which qualitative and quantitative methods are combined. A valid and practicable measurement instrument (“UFair” University Fairness Questionnaire) will be developed and tested, and the relationship with various health outcomes will be examined. The UFair questionnaire will be made available free of charge to other researchers.

## 1. Introduction

Distress is a common phenomenon in today’s society with serious health consequences. Chronic stress exposure is related to increased depressive symptomatology, burnout syndromes, and sleep disturbances [1]. University students face a variety of stressors and there is growing evidence that distress during university presents an increasing problem and is associated with poorer learning outcomes, and lower quality of life, as well as poorer mental health, resulting in depression, anxiety, and burnout [2,3,4,5,6].

Preceding studies on the assessment of students’ stress levels mostly used survey instruments that measure general feelings of stress. However, these appear not suitable to inform about the direct stress load caused by studying or to derive concrete preventive measures [7,8]. Furthermore, these studies often lack a theoretical foundation and a direct relation to the target group of students. Therefore, it is important to survey students’ direct stress from studying in a theory-based manner in order to develop suitable and effective prevention measures. This is important to ensure that students can transition into their working lives as healthy as possible.

A promising starting point is offered by established psychosocial stress models from the occupational setting, such as the Effort-Reward-Imbalance (ERI) model of Siegrist [9]. The theoretical foundation of the ERI model bases on the principle of social reciprocity, a fundamental norm of interpersonal behavior. Social reciprocity is characterized by the expectation of receiving an equivalent reward for the effort spend. If the reward is less than the provided effort, negative emotions and distress arise, which can lead to chronic stress reactions and negative health consequences. In the occupational setting, the ERI model received ample empirical evidence [10,11,12,13]. An ERI-Questionnaire was recently adapted to the target group of university students (ERI-Student Questionnaire) [14] and tested with nearly 690 students from different disciplines at German universities [15]. All theoretically predicted components of the model, such as high effort in combination with low reward (the so-called effort-reward-imbalance) were significantly related to both poorer self-reported health and symptoms of depression and anxiety [15]. Thus, it has been successfully demonstrated that the ERI model, which was originally applied in work stress research, can also be transferred to the university context and can show the expected associations in this setting.

In the planned study, this research will be pursued, as another theoretical stress model, which is established in the work context and complementary to the ERI model, also appears promising in the university context: the organizational justice model [16]. Perceived organizational justice is a stress model that captures the extent of unfair treatment in the workplace. Although there is some overlap between the ERI model and the organizational justice model, there are also important differences. While the ERI model addresses the intrapersonal justice of exchange based on distinct labor market conditions and is related to a contractual exchange, the organizational justice model addresses interpersonal inequity in fair allocation of scarce resources [17,18]. Apart from theoretical aspects, empirical findings also confirm the complementarity of the two models (e.g., [18,19]). The organizational justice model thus has the potential to shed light on other stressful psychosocial issues in university that have been little considered and researched. According to Byrne and Cropanzano (2001) [20], organizational justice can be defined broadly as perceptions of fairness in the workplace. The absence of such perceived fairness in the workplace has been shown to be strongly related to various health indicators [21,22].

Several theoretical explanations exist for why people’s experience of injustice is associated with increased distress. The basis consideration is that humans have developed an evolutionary sense of fairness to enable long-term cooperation, which can also be found in primates [23]. In line with this deontic model of justice [24], the idea is—similar to the ERI model—that perceived injustice represents a violation of a social norm (a fairness norm) and is therefore associated with negative emotions [25]. Another explanation assumes that perceived justice enables the achievement of personal goals by making them predictable and controllable [26]. Other approaches view a lack of fairness as a source of distress because it can be seen as a sign to employees that their job demands exceed their individual coping resources [27]. Similarly, the injustice stress theory [28] states that unfair treatment is primarily a stressor because it reveals the gap between one’s capabilities and the demands from the work environment. Even though there is no consensus about why perceived unfairness causes distress, the empirical findings for this relationship are incontrovertible. Empirical research has shown a robust association of organizational justice with various stress indicators and health outcomes cross-sectionally and longitudinally. These include, for example, psychological and physiological stress [29,30,31,32], mental health [33,34,35,36], cardiovascular disease [37,38,39,40], and health-related risk behaviors [41,42].

The model of organizational justice consists of three components (see Table 1). The three components are, first, distributive justice, which refers to the fair distribution of benefits and outcomes [43], second, procedural justice, which refers to justice in general processes and procedures [44], and third, interactional justice, which refers to fair treatment in interpersonal interactions [45]. Each of these components has its own rules, which are listed in Table 1.

Similar to the ERI model, organizational justice can be assumed to have its eligibility not only in the workplace, but also in the university context. A recent study of 543 Spanish students showed that organizational justice was related to engagement (r = 0.88) and burnout (r = 0.79) [46]. Thus, similar to the violation of the norm of social reciprocity (captured via the ERI student questionnaire), perceived inequity in the university context might also pose a health risk. However, a feasible and valid measurement tool has been lacking to date. The study from Spain used a questionnaire for its survey, which was developed and designed specifically for the workplace [46]. A specific measurement tool, adapted to the university context, would be able to more accurately and validly measure the stress of students and would be able to more precisely show the relationship with health indicators. This would enable the derivation of a healthy environment conducive to learning for students and the elimination of potential sources of poor health at an early stage.

For these reasons, the planned project will transfer the construct of organizational justice to the university context by means of established scientific procedures and adapt it to the target group. A valid and practical survey instrument (“UFair” University Fairness Questionnaire) will be developed, which records the respective degree of risk for injustice in the university context as well as its relation with various health outcomes among the target group of university students. For this purpose, qualitative and quantitative methods (a so-called mixed-methods research design) will be used.

## 2. Experimental Design

A mixed methods study is planned to complement established quantitative methods with qualitative methods in order to bring together the advantages and strengths of both approaches [47]. In this project, the combination of qualitative and quantitative methods offers significant added value compared to exclusively using quantitative methods. Qualitative methods can identify fundamental aspects inductively, which can then be tested quantitatively in a hypothesis-driven manner. In this case, it is possible to develop an instrument for the quantitative survey through direct interactions with the target group. A purely quantitative orientation might lead to the development of items that do not cover the students’ concerns and perspectives [48]. Thus, it might miss the target group, since important aspects may not be recorded or might be overlooked. Therefore, it is considered extremely beneficial to use innovative methods that already include the perspective of those affected during the development of the instruments. In this way, qualitative methods start directly at the source—i.e., the students themselves—without the risk of bias caused by the specifications of the researchers conducting the survey (e.g., through prefabricated, missing items, misleading formulations, or limited response categories). In this way, relevant determinants that will be later examined quantitatively in depth are generated by those who are directly affected by the problem and generated items can therefore provide the best possible insights.

In the planned study, it is reasonable to start with the qualitative part of the study in order to identify aspects of organizational justice in the university context among the students in an open and unbiased way. The qualitative part will be followed by a quantitative written pretest and a nationwide quantitative online survey among students. Thus, the project follows the exploratory sequential design [47,49]. An outline of the study parts is presented in Figure 1.

## 3. Detailed Procedure

The study process of the development of the UFair Questionnaire (University Fairness) can be integrated into the seven-step process for questionnaire development, the AMEE Guide No. 87 [50], based on preliminary work by Gehlbach et al. (2010) [51]. The qualitative interviews form step 2, the written pretest step 6 and the nationwide survey step 7 (see Figure 2).

### 3.1. Step 1: Review of Literature and Creation of Preliminary Items

To gain a comprehensive insight, previously existing questionnaires for surveying organizational justice will be compiled to derive relevant items for the university context. Based on this, a list of potential items will be developed in a next step. If items are available in English, a five-step back-translation will be carried out [52].

This list of items will be supplemented by items formulated by the project team based on the above-mentioned justice rules in Table 1. These rules are intended to serve as a basis for defining perceived (in)justice in the university context.

The items will be classified according to the four categories of the taxonomy for justice scales developed by Colquitt and Shaw (2005) [53]. The first category relates to the components of justice: is distributive, procedural, or interactional justice measured (see Table 1)? The second category relates to the source of justice: is it a human decision-maker or a formal organizational system? The third category is about the context of justice: does the (in)justice relate to a single, specific event or is it characterized by persisting circumstances characterized by multiple events? The fourth category concerns the measurement approach: is it a direct assessment of the perception of justice (“how fair is …”) or an indirect assessment, for example through items focusing on the justice rules (e.g., are procedures consistent across time and persons). This taxonomy will be used to ensure that all areas (i.e., components, sources, contexts, and measurement approaches) will be covered.

Based on this theoretical foundation, in the following step two, the students will be involved through qualitative interviews in the (further) development of the concept of organizational justice in the university context. The students’ living environment and interactions with teachers and fellow students regarding their perception of university justice will thus be mapped. From the perspective of scientific theory, this connects theory-based to empirical work in a different lifeworld with the practical and applied empirical evidence of the population of interest.

### 3.2. Step 2: Conducting Qualitative Interviews: Explorative-Inductive and Translational-Deductive

The first exclusively deductive step—inferring from existing concepts and academic work—will be followed by an inductive step. In this way, the students’ ideas of justice will be captured in an explorative way (part one of the qualitative interviews). Following an interview guideline, the students freely report on their experiences of justice in the university context. At the same time, in addition to this explorative procedure, the concept of university justice will also be discussed with the students (part two of the qualitative interviews). In this way, it can be deductively examined to what extent the concept of organizational justice plays a role for students and how it is shaped in the university context.

The qualitative guided interviews with 20 students have two aims. First, they will be used to learn more about how students conceptualize and describe the concept of justice, to what extent it is seen as relevant and existing, what they see as (possible) sources of injustice in the university context, and how they usually react when experiencing injustice. This can uncover previously unnoticed unfair facts and will supplement the item list in the next step. Secondly, the items developed in step 1 will be discussed with the students and critically debated. The general background information on individual items and the perception among the students, as well as the perceived importance will be recorded.

The students for the qualitative interviews will be recruited at structurally very different universities and to achieve the most comprehensive knowledge of the aspects of unfairness at universities, a heterogeneous spectrum of students will be obtained (in terms of age, gender, number of semesters, field of studies, type of university, etc.). All students will receive a voucher of 20€ for participating in the interview. Interviews will be audiotaped and transcribed verbatim.

The qualitatively collected data will be evaluated by means of a content analysis based on Mayring [54,55]. In this way, it will be possible to work out different themes and categories with regard to the perception of (in)justice. This will be done both from the material (inductive) and based on prior theoretical knowledge (deductive).

### 3.3. Step 3: Synthesis of the Qualitative Results with the Literature

After the evaluation of the qualitative data, the students’ ideas of (in)justice will be analyzed in more depth. It is important to consider the student’s context. A synthesis of the qualitative results of the construct of university justice and the theory will thus be possible. In addition, the theoretical meaningfulness of transferring the concept of organizational justice to the university context will also be examined. At that, the following questions are central: Do students perceive organizational injustice in the university context? To what extent does justice influence their actions, interactions, and subjective well-being? To what extent can the dimensions of organizational justice be transferred to the university setting? What is the degree of congruence between the theoretical construct (i.e., the deductive groundwork) and the students’ subjective perceptions? Based on the answers to these questions, the questionnaire items to be tested will be formulated in the following step.

### 3.4. Step 4: Development of the Questionnaire Items

In this step, the preliminary items are formulated based on the synthesis described in step 3. New items will be formulated based on the qualitative findings and the pre-formulated items will be adapted based on theory and questionnaires from the literature in order to best reflect the university context—and thus the students’ living environment. In addition, it must be ensured that a language is used that is understood by the students.

### 3.5. Step 5: Validation by Experts

To check the content validity, the developed items will undergo an expert review. For this purpose, experts in the field will meet and critically review the item battery created. The focus will be on checking clarity, relevance, and completeness.

### 3.6. Step 6: Pretest of the Questionnaire Items

This first version of the UFair Questionnaire will then be pretested with 30 students. The students will answer the items, which enables initial statistical evaluations. In addition, they should indicate which items are incomprehensible or misleading, and they can suggest different wordings. Within the framework of this pretest, cognitive pretest techniques will be used on a written basis: testing the comprehensibility of formulations (comprehension probing), optimizing answer categories (category selection probing), testing numerical data (information retrieval probing), and testing the reliability of answers (confidence rating). Based on this written pretest, a first analysis of the internal consistency (reliability: Cronbach’s alpha and item-total correlation) will be carried out. Furthermore, the correlative relationship with relevant constructs, such as general stress perception and self-reported health will be estimated (criterion validity). Finally, the first version of the UFair Questionnaire will be revised and further improved based on the results of the pretest.

### 3.7. Step 7: Piloting the Questionnaire

The final step is the piloting of the UFair questionnaire in the target group. For this purpose, a nationwide quantitative online survey will be conducted in order to comprehensively test the psychometric properties of the UFair questionnaire. Classical test theory and item response theory will be used to obtain a measuring instrument that is as valid and time efficient as possible. During the classical test theory, the reliability of the items will be checked by means of item-total correlation and Cronbach’s alpha (internal consistency). The construct validity will be tested with an exploratory factor analysis, which is supplemented by a confirmatory factor analysis using structural equation modeling. Subsequently, regression models will test criterion validity by examining the relationship of the UFair questionnaire with theoretically and empirically related constructs, such as self-reported health, general stress experience, and mental health. Finally, probabilistic procedures (Rasch models) will be used to estimate item difficulty and the ability of a person to identify the best items and to obtain the shortest and most efficient questionnaire possible as a final product. In order to be able to carry out a comprehensive test, in addition to the UFair Questionnaire and scales for testing criterion validity—such as subjective health status, mental health, and general stress experience—possible confounding factors, such as socio-demographics (including age and gender), and health-related risk behavior (including alcohol consumption, tobacco consumption, sleep behavior) will also be assessed in the online survey.

It is planned to survey 1000 students that represent the actual distribution of students by gender, field of study, and type of higher education institution (65% universities and 35% universities of applied sciences [56]) in Germany. Furthermore, the sample will be divided equally into participants who complete the questions via mobile devices (smartphones, tablets, *n* = 500) and via fixed devices (*n* = 500) in order to counteract possible survey effects and to increase objectivity. The online survey will be conducted by an external market research institute.

The sample size of 500 is set for each type of survey, even though a sample size calculation based on the results of the Spanish study considers a significantly lower number of persons to be sufficient. The Spanish study reports a medium-strong negative association of organizational justice with burnout (beta = −0.51, SD = 0.14, R^2^ = 0.24) [46]. Based on this, the required sample size would be 39 (96) individuals, with a power of 0.95 (0.99) and a significance level of α = 0.05 (0.001) (G*Power, version 3.1.9.4; t tests—Linear bivariate regression: One group, size of slope, two-tails [57]). However, this estimate is based on a bivariate relationship with one group. In this study, a higher power is needed since we plan to control for confounding factors (e.g., lifestyle factors, number of semesters, etc.) to stratify the sample (e.g., by type of university, gender, survey type, field of study) and to conduct different psychometric tests. Recommendations and common sample sizes of other validation studies ranged from at least 300 to approximately 1500 persons [58,59]. Accordingly, in this project a sample size of 1000 (2 × 500 students per survey type) should be realized.

## 4. Expected Results

The aimed result of this project is the successful adaptation of the concept of organizational justice to the target group of students. By applying a mixed methods approach and involving the target group, it will be ensured that the developed UFair questionnaire is understood and measures what was conceptually intended. It is planned to make the final UFair questionnaire freely available on the Institute’s homepage so that other researchers can use it free of charge and apply it in their studies. The evaluation of the measurement instrument can make it possible to create a healthier environment for students that is more conducive to learning and to identify potential stress sources at an early stage and subsequently eliminate them. On a theoretical level, this study has the potential to gain a better understanding of justice perceptions in the population of university students, adapt the organizational justice model for this population, and develop a specific theoretical model. For example, it is conceivable that a new justice dimension is relevant in this context and must be considered. Further studies might test the developed model in other countries and other cultural contexts with a different education system, as this project is aligned with the German system.

## 5. Conclusions

Assessing fairness in the university context may help to develop or improve supporting structures for students similar to already existing counseling opportunities at the university. Spreading awareness of the potential existence of organizational unfairness would be helpful for universities, faculty staff, and teachers, who may not have heard about this concept before. Supporting students who perceive themselves to be treated unfairly might help to forestall the development of mental illness as well as educational aspects, such as poor learning outcomes and dropping out. This research may also influence and inspire further educational injustice research in a similar and related area, such as classroom justice—the relevance of perceived fairness of the teachers’ assessment practices on performance (e.g., [60,61,62])—or efforts to implement artificial intelligence fairly in education [63].

## Figures and Tables

**Figure 1 ijerph-19-16340-f001:**
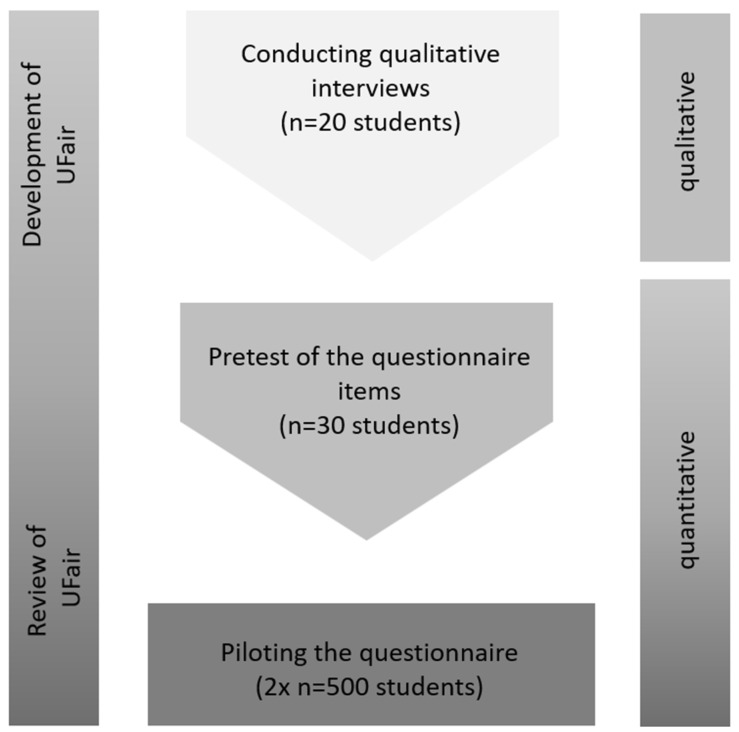
Outline of the study course.

**Figure 2 ijerph-19-16340-f002:**
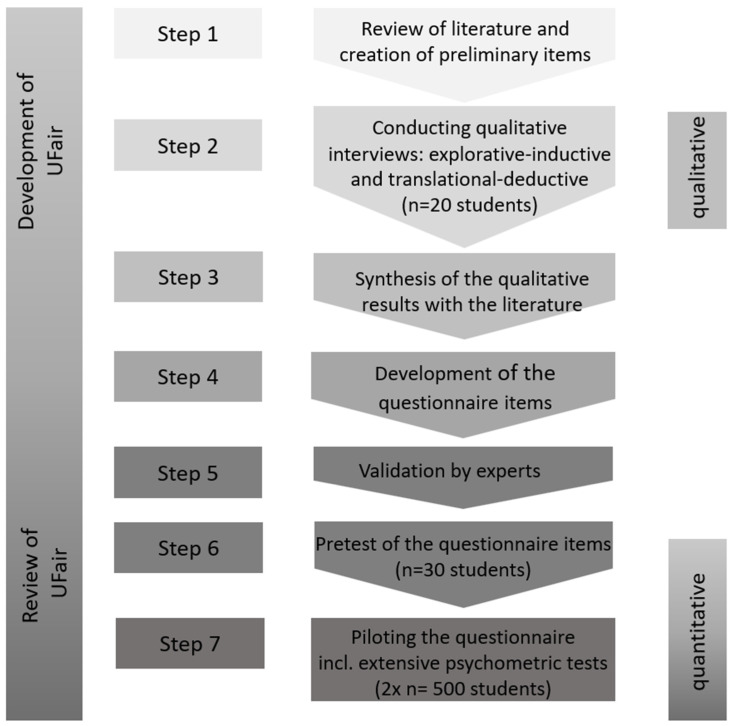
Steps of the questionnaire development UFair following Artino et al. (2014) [50].

**Table 1 ijerph-19-16340-t001:** Overview of the three justice components with the associated rules.

Justice Components
Distributive justice: fairness of distribution of outcomes 1. Equity: in accordance with contributions
2. Equality: each individual the same
3. Need: in accordance with the most urgency
Procedural justice: fairness of the decision-making process leading to the allocation of outcomes
1. Consistency: consistency of the procedures across time and employees
2. Bias-suppression: absence of personal bias and favoritism
3. Accuracy: decisions based on good and as much as possible information
4. Correctability: modification or reversibility of decisions
5. Representativeness: consideration of the interests of those affected by decisions
6. Ethicality: allocation procedures should be compatible to ethical standards
Interactional justice: fairness in interpersonal interactions
1. Truthfulness: open, honest, and candid
2. Respect: treat individuals with sincerity and dignity
3. Justification: providing adequate explanations
4. Propriety: no prejudicial statements or improper questions

## Data Availability

Not applicable.

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
