# Peer review of "University Fairness Questionnaire (UFair): Development and Validation of a German Questionnaire to Assess University Justice—A Study Protocol of a Mixed Methods Study"

_ijerph, 2022, doi:10.3390/ijerph192316340_

Round 1

Reviewer 1 Report

It is suggested that:

- The introduction should exhibit in detail how both the ERI and the organizational justice models are complementary or support the first one to the second one.

- Referring to the interviews, it is suggested to consider an increase in the number of them considering the population.

- Respect to the sample size on piloting the survey, it is suggested to increase the sample size in consideration of the costs on applying the online survey.

Reviewer 2 Report

The chosen research topic is very interesting, given its importance and transcendence at a global level. The title is excessively long and others could be used that respond better to its content, for example, “University Equity Questionnaire based on the Concept of Organizational Justice”. The goal is explicitly stated. Its structure is adequate. The writing is fluid, with a defined plot line and clear exposition. The theoretical argument is adequate. The quantity and quality of the documentary sources used is appropriate. The limitation of this contribution lies in not specifying the aforementioned questionnaire which, together with some deficient conclusions, make the contribution of the research very limited for a journal of this category.

I would advise the authors to review the articles that specify the proposed model, for example, https://doi.org/10.3390/su11174639

1. GRI (Global Reporting Initiative). G4 Guía para la elaboración de memorias de sostenibilidad. Principios y contenidos básicos. Available online: http://www.mas-business.com/docs/Spanish-G4.pdf (accessed on 6 June 2019). 2. ISO 26000. Guidance on Social Responsibility; Business Expert Press: Geneva, Switzerland, 2017. [Google Scholar] 3. COM. Libro verde. Fomentar un marco europeo para la responsabilidad social de las empresas. 2001. Available online: http://www.europarl.europa.eu/meetdocs/committees/deve/20020122/com(2001)366_es.pdf (accessed on 6 June 2019). 4. OIT. Pautas de la OIT sobre trabajo Decente y turismo Socialmente Responsable; Oficina Internacional del Trabajo: Ginebra, Switzerland, 2017. Available online: https://www.ilo.org/wcmsp5/groups/public/---ed_dialogue/---sector/documents/normativeinstrument/wcms_546341.pdf (accessed on 6 June 2019). 5. UNWTO. Global Code of Ethics. 1999. Available online: http://ethics.unwto.org/content/global-code-ethics-tourism/ (accessed on 6 June 2019). 6. McWilliams, A.; Siegel, D.S.; Wright, P.M. Corporate social responsibility: Strategic implications. J. Manag. Stud. 2006, 43, 1–18. [Google Scholar] [CrossRef] 7. Kim, H.; Hur, W.; Yeo, J. Corporate brand trust as a mediator in the relationship between consumer perception of CSR, corporate hypocrisy, and corporate reputation. Sustainability 2015, 7, 3683–3694. [Google Scholar] [CrossRef] 8. Pérez-Aranda, J.A. Valoración de la Resposabilidad Social Empresarial (RSE) Por La Demanda Hotelera. Ph.D. Thesis, Universitat Jaume I, Castellón de la Plana, Spain, 1 September 2016. Available online: http://dx.doi.org/10.6035/40012.2016.384009 (accessed on 6 June 2019). 9. Boronat-Navarro, M.; Pérez-Aranda, J.A. Consumers’ perceived corporate social responsibility evaluation and support: The moderating role of consumer information. Tour. Econ. 2018, 25, 613–638. [Google Scholar] [CrossRef] 10. Bigné, E.; Chumpitaz, R.; Andreu, L.; Swaen, V. Percepción de la responsabilidad social corporativa un análisis cross-cultural. Univ. Bus. Rev. 2005, 5, 14–27. [Google Scholar] 11. Petrović-Ranđelovića, M.; Stevanovićb, T.; Ivanović-Đukićc, M. Impact of corporate social responsibility on the competitiveness of multinational corporations. Proced. Econ. Financ. 2015, 19, 332–341. [Google Scholar] [CrossRef] 12. Shin, I.; Hur, W.; Kang, S. Employees’ perceptions of corporate social responsibility and job performance: A sequential mediation model. Sustainability 2016, 8, 493. [Google Scholar] [CrossRef] 13. Ley 2/2011, de 4 de marzo, de Economía Sostenible. 2011. Available online: https://www.boe.es/buscar/pdf/2011/BOE-A-2011-4117-consolidado.pdf (accessed on 6 June 2019). 14. Carroll, A.B. A Three Dimensional Conceptual Model of Corporate Social Performance. Acad. Manag. Rev. 1979, 4, 497–505. [Google Scholar] [CrossRef] 15. Simon, D. Sustainable development: Theoretical construction or attainable goal? Environ. Conserv. 1989, 16, 41–48. [Google Scholar] [CrossRef] 16. Šlaus, I.; Jacobs, G. Human capital and sustainability. Sustainability 2011, 3, 97–154. [Google Scholar] [CrossRef] 17. eronen, E. Sustainability and Sustainable Development. In Encyclopedia of Corporate Social Responsibility; Idowu, S.O., Capaldi, N., Zu, L., Gupta, A.D., Eds.; Springer: Berlin/Heidelberg, Germany, 2013. [Google Scholar] 18. Bell, S.; Morse, S.; Lockie, S. Sustainability Indicators: Measuring the Immeasurable. Impact Assess. Proj. Apprais. 2001, 19, 171. [Google Scholar] ...

Reviewer 3 Report

This paper dealt with the creation of a questionnaire to measure the extent to which students perceive unfairness in the university context. The authors adopted a mixed methods study that led to a questionnaire including both quantitative and qualitative perspectives. Throughout the paper, the process  the authors plan to use for this purpose was described, by splitting it into seven steps. Short conclusions were finally provided. 

Overall, the topic of this paper is really relevant and prominent - it is really important research! The paper was generally easy to follow, with often well-motivated choices. A merit of this work is that the authors focused on both qualitative and quantitative perspectives, which would allow them to collect opinions and feelings in multiple formats and from multiple sources, given also flexibility to the students. Despite these positive aspects, the reviewer was left with several open questions during the reading and believes that the study is not yet mature for being published as a journal-archiving publication. As far as it is presented, the content would better fit as a workshop or short paper at a conference, where authors could benefit from feedback by other researchers to improve the questionnaire and the process, before running it at scale. Once the large scale study would be conducted, the study could be extended to a journal publication. In what follows, I will detail a few elements that should be addressed in future versions of the work, along with some suggestions.

First, though the introduction presents prior work strictly related to the presented study, it remains unclear what are the concrete novel contributions of this paper. This aspect should be clarified in the introduction - it is often done through a bullet list with the two / three key novel contributions. From the reviewer's perspective, the novel contribution at this stage is merely the definition of a process to create the questionnaire, while the questionnaire itself and the findings from the questionnaire's answers could not be arguably considered as a contribution, given that the exact questions seem not to be reported in the paper (and therefore could not be judged by the reviewers nor used in the future by other researchers) and there is no presentation of results on a reasonably large study with students (though it is planned) - to see whether they found the questionnaire useful and what has been found about fairness in those specific contexts. The clear suggestion is to provide and clearly explain the questionnaire's content in the paper. Another suggestion is that, given that the contribution would be questionnaire, there should be an evaluation component (including metric scores and not only the process description) to assess explictly whether the selected questions in the questionnaire were found relevant by students.

Second, the design and methodology appear at a very early stage where they are planned but not delivered to the target students. This consideration questions the extent to which the proposed methodology will work or require changes (and so updates on the method and the paper) once it will be run with students. To strenghten the paper from this perspective, I invite the authors to move forward with their study, including the steps mentioned in the paper (e.g., validation with experts) but also by selecting one or two universities as case studies and providing the questionnaire to the respective students. To be self-contained and allow the community to benefit from it, the methodology described in the paper should be validated and refined according to these pilots (to avoid having obsolete practices or practices that finally did not work in the final version of the paper) and the paper should provide a clear description of the results obtained with the two case studies. I suggest that these case studies should aim to show to the reader what can be found from the questionnaire and how the answers can be analyzed in the context of this new questionnaire.

Third, once the results from the two case studies have been obtained, I suggest that the paper should include a final discussion section where the authors collect all the findings from their case studies and link them to those from other domains (e.g., the questionnaire provided in the workplace by [45]). In this way commonalities and dissimilarities across domain could be highlighted, representing another added value of this study. To improve the discussion, I also invite the authors to frame the paper according to two or three research questions and organize the results section according to them (e.g., one subsection for each research question). Beyond the questionnaire, I believe that the findings from the case studies would be another key novel contribution that will help the community shape the next steps in this area.   

As a minor suggestion, I invite the authors to contextualize their work more broadly with other questionnaires developed to assess fairness in the context of education. By scanning the main paper libraries, as an example, another study studied the differences between students and teachers in the perception of fairness (Sonnleitner, P., & Kovacs, C. (2020, February). Differences between students’ and teachers’ fairness perceptions: Exploring the potential of a self-administered questionnaire to improve teachers’ assessment practices. In Frontiers in Education (Vol. 5, p. 17). Frontiers Media SA.); another work assessed fairness in education through a questionnaire but from the perspective of the adopted AI tools (Fenu, G., Galici, R., & Marras, M. (2022). Experts’ View on Challenges and Needs for Fairness in Artificial Intelligence for Education. In International Conference on Artificial Intelligence in Education (pp. 243-255). Springer, Cham.). Highlighting (dis)similarities with such works, to be identified with another run of literature search, would let the community better appreciate the novelty of this study.

Reviewer 4 Report

Dear Editor

The subject matter is very interesting and innovative. The effort made by the authors is certainly worthwhile. Although it is an effort that has been initiated and concluded. The problems and justification of the research are well described and substantiated. I fully agree with the authors. My acknowledgement once again. As well as the phases of the research and the construction and validation of results. However, no conclusive results are presented, which, despite the interest of the research process as a whole, reduces the interest of the article itself. I believe that this point needs to be improved, not only for the approval by the journal editors but also for the interest of its reading audience. For example, although it seems that the experimental validation of the instrument is not done, the results of the interviews and/or the pre-validation with 30 subjects could have been included. Supporting both in the statistical rationale that covers them. I would strongly urge you to include some results to increase the interest of the article. 

Best regards

Round 2

Reviewer 2 Report

In this research, the conclusions provided are still limited because, although the practical implications are revealed, no mention is made of their contribution at a theoretical level. The limitations of the study and future lines of research must also be pointed out.

Reviewer 4 Report

Dear editors,

The authors refuse to present results, which are the bulk of a research article. Without them, the text lacks interest for readers. Only proposing a research topic and your ongoing or forthcoming research project is appropriate for applying for funding and recognition for a research project. Not for publication, which lacks interest as presented.

Best regards
